# A mixed methods study to develop a tool to assess institutional readiness to conduct knowledge translation activities in low-income and middle-income countries

Anna Kalbarczyk  , Aditi Rao, Olakunle Alonge

International Health, Johns Hopkins University Bloomberg School of Public Health, Baltimore, Maryland, USA

**Correspondence to**
Dr Anna Kalbarczyk;
akalbarc@jhu.edu

## ABSTRACT

**Objective** This paper describes the development of a tool for assessing organisational readiness to conduct knowledge translation (KT) among academic institutions in low-income and middle-income countries (LMICs).

**Design** A literature review and stakeholder consultation process were conducted to identify constructs relevant for assessing KT readiness in LMICs. These were face-validated with LMIC stakeholders and organised into a Likert-scale questionnaire.

**Participants** The questionnaire was distributed to researchers based at six LMIC academic institutions and members of a global knowledge-to-action thematic working group.

**Outcome measures** An exploratory factor analysis was used to identify underlying dimensions for assessing institutional readiness to conduct KT.

**Results** 111 respondents with varied KT experiences from 10 LMICs were included in the analysis. We selected 5 factors and 23 items, with factor loadings from 0.40 to 0.77. These factors include (1) institutional climate, (2) organisation change efficacy, (3) prioritisation and cosmopolitanism, (4) self-efficacy, and (5) financial resources. These factors accounted for 69% of the total variance, with Cronbach's alpha coefficients of 0.78, 0.73, 0.62, 0.68 and 0.52, respectively.

**Conclusions** This study identifies a tool for assessing readiness of LMIC academic institutions to conduct KT and unique opportunities for building capacity. The organisational focus of these factors underscores the need for strategies that address organisational systems and structures in addition to individual skills. Future research will be conducted to understand determinants of these factors and develop a comprehensive set of capacity building strategies responsive to academic institutions in LMICs.

## STRENGTHS AND LIMITATIONS OF THIS STUDY

⇒ The development of this tool was rooted in organisational change theory, building on validated tools.
⇒ We conducted the research across multiple contexts in both Africa and Asia to capture constructs generalisable to institutions based in low-income and middle-income countries.
⇒ A significant limitation of this research is the small sample size; however, studies have argued smaller sample sizes may be justified when higher correlation coefficients are observed.
⇒ The original tool was lengthy, which could have resulted in response fatigue, leading to inaccurate data or incomplete responses.

Cochrane review identified a wide range of strategies for implementing such interventions in LMICs.[3] This gap between what we know through research and what is done in practice (ie, 'know-do' gap) underscores the need for improved utilisation of research by policy makers and practitioners in LMICs.

Knowledge translation (KT) is one approach to addressing this gap that describes a process of research generation, synthesis, sharing, and utilisation to improve health. Many definitions for KT exist and we use the one developed by the Canadian Institutes of Health Research: 'A dynamic and iterative process that includes the synthesis, dissemination, exchange, and ethically sound application of knowledge to improve health, provide more effective health services, and strengthen the health care system'.[4] Examples of KT activities include evidence synthesis via policy briefs and systematic reviews, sharing knowledge through communities of practice, and forming large-scale partnerships and networks, which can be applied at different policy and practice levels to facilitate uptake of evidence and improve population health.[5]

## INTRODUCTION

Evidence-based, cost-effective interventions designed to improve health are available but underutilised, particularly among the poorest populations in low-income and middle-income countries (LMICs).[1] [2] A recent

Academic institutions in LMICs are well positioned to conduct KT activities as respected producers of research evidence and their ability to serve as knowledge brokers, but face a unique set of barriers driven by contextual factors including structural inequities (eg, historically imbalanced relationships with institutions in the Global North and research agendas driven by outside researchers)[6 7] and limited resources.[8] While some of these contextual factors (eg, structural inequities) may be generalisable across most LMICs, others are specific to countries and may even vary within countries, for example, the role of leadership, organisational structure and culture, and the broader sociopolitical context. We conducted a qualitative study with participants in academic institutions in six countries (Bangladesh, Democratic Republic of the Congo (DRC), Ethiopia, India, Indonesia and Nigeria) to identify barriers and determinants of institutional capacity to conduct KT activities across diverse LMICs contexts, and found that soft-skills (ie, communication, self-awareness and adaptability), robust networks, and alignment between institutional priorities and incentives are important factors that shape institutional capacity to conduct KT activities in LMICs.[9] While strategies such as trainings, mentorship and institutional leadership engagement have been developed to address these barriers and determinants,[10 11] these strategies have been mostly applied to academic institutions in high-income countries (HICs),[12–14] with limited empirical evidence of their effectiveness in LMIC settings.[15] Irrespective of the strategies enacted to address barriers to conducting KT activities, academic institutions need to be self-aware of their role in KT and their readiness to undertake relevant KT activities addressing priority health issues to be successful.[16]

Indeed, some institutions are becoming increasingly aware of the importance of organisational readiness and the need to implement processes that motivate and enhance change to facilitate the conduct of KT activities or address barriers to conducting KT activities.[17 18] Such processes could be motivated by increased demand from policy makers for evidence-based data or funders offering new opportunities that prioritise or require KT activities. Whereas specific KT activities (eg, systematic review, establishing a community of practice) may vary in different academic institutions given differences in their priority health issue focus, policy cycle and resources, the similarities in the structure, function and objectives of academic institutions provide a conceptual basis for assessing their readiness to conduct different KT activities across different settings. Assessing organisational readiness enables exploration of facilitators and barriers for individuals and organisations to implement capacity building strategies,[19] and has been used to tailor approaches for addressing barriers to KT activities.[20] However, most of these readiness assessments have been conducted in HICs, and there is an underuse of validated tools.[18]

Validated readiness tools, used to assess institutional readiness for change (eg, to implement any intervention, including KT activities), include the Organizational Readiness for Implementing Change,[19] Organizational Readiness to Change Assessment[21] and Texas Christian University Organizational Readiness to Change (TCU ORC).[22] These tools largely overlap and assess domains such as availability of resources, individual attributes and motivation, and organisational climate.[19 21 22] Psychometric assessments have been conducted on these tools which largely support their reliability and structure.[19 23 24] Most tools have been implemented in HICs, largely in healthcare settings including hospitals, primary care settings, public health agencies and public sector organisations, and not specifically for KT. Gagnon *et al* [20] developed the Organizational Readiness for Knowledge Translation (OR4KT) tool, designed to assess organisational readiness for KT in healthcare organisations, but they note that its applicability to LMICs may be limited given how different healthcare organisations are organised comparing HIC with LMIC settings. Our literature review identified only one validated tool, the TCU ORC, that had been used in an LMIC.[25] This study used the TCU model to examine the organisational functioning of drug treatment facilities in South Africa but makes no mention of adapting the tool to address unique contextual factors. A recent systematic review identified 30 tools designed to address organisational readiness for institutions in LMICs but none were validated.[26] The lack of readiness assessments specific to KT and availability of adapted tools for conducting readiness assessment further contributes to the 'know-do' gap in LMICs.

This paper describes an empirical study to develop a tool for assessing institutional readiness to conduct KT among academic institutions in six LMICs (Bangladesh, DRC, Ethiopia, India, Indonesia and Nigeria). The paper builds on published tools for readiness assessments more broadly and developed for HIC settings and identifies constructs that are relevant for assessing readiness of academic institutions to do KT activities in LMICs, and the validity and reliability of an adapted tool measuring these constructs for LMIC contexts.

## METHODS

This research is embedded in a 5-year parent project, 'Synthesis and Translation of Research and Innovations from Polio Eradication' (STRIPE), designed to map, synthesise and disseminate knowledge from global efforts to eradicate polio.[27] The STRIPE consortium is led by Johns Hopkins University in collaboration with six academic institutions and one research consultancy firm based in seven LMICs. This research engaged the six academic collaborators, shown in table 1, and was conducted in three sequential steps.

### Step 1

A *literature review* was conducted to identify validated tools for measuring individual and institutional readiness. Standard search tools including PubMed and Google

**Table 1** STRIPE academic partners

| Institution name | Country |
| --- | --- |
| James P Grant School of Public Health, BRAC University | Bangladesh |
| School of Public Health, University of Kinshasa | Democratic Republic of the Congo |
| College of Health Sciences, Addis Ababa University | Ethiopia |
| Indian Institute of Health Management Research University | India |
| Faculty of Medicine, Public Health, and Nursing, Gadjah Mada University | Indonesia |
| College of Medicine, University of Ibadan | Nigeria |

STRIPE, Synthesis and Translation of Research and Innovations from Polio Eradication.

Scholar were searched in 2019 using combinations of the following terms: 'tools', 'organizational readiness' and 'institutional assessments'. Thematic constructs and questionnaire items from the identified tools were extracted and organised by common domains. These were then mapped to the domains and constructs of the Consolidated Framework for Implementation Research (CFIR), which helped identify important gaps for implementation in LMICs. CFIR is a well-known framework, comprised of five domains, namely intervention characteristics, outer setting, inner setting, individual characteristics and implementation process.[28] The framework has been widely used in LMICs as a practical structure for approaching complex programmatic and policy challenges (from planning to implementation to evaluation) such as conducting KT activities.[29]

Institutional data were also collected from each academic partner, including strategic plans, organograms, and information on recently published articles. Public information from each country's Ministry of Health (MoH), including recent reports and strategic plans, was collected. Using the institutional data and information from the MoH, a content analysis was performed to establish health reference points for participants from each country (ie, specific priority health issues on which the academic institution and the MoH align).

### Step 2
We conducted *stakeholder interviews* to evaluate the face and content validity of the constructs and items described from the review phase for LMIC contexts and explore new themes. The 18 stakeholders interviewed included representatives from each of the six academic STRIPE institutions in addition to their external government partners. Internal representatives met at least one of the following criteria: (1) individuals involved in institutionally prioritised KT activities (health issue-specific), (2) individuals involved in making strategic/policy decisions around KT activities at the institutional level, and (3) individuals in leadership that determine the internal context and external relationships of the institutions. External stakeholders included policy makers who currently engaged or had been engaged with the academic institution in the past 2 years in efforts to conduct KT activities.

Inputs from this process focused largely on providing feedback on constructs of readiness as they related to less well-operationalised domains and constructs of CFIR in the literature (eg, outer setting and characteristics of individuals). This process was also used to confirm health priorities in each country, identified initially in the literature review, and develop a list of KT activities commonly conducted across the settings. A draft tool for readiness assessment was then developed based on findings from the review and consultation steps. Country-level priorities and activities were incorporated throughout the tool to help illustrate complex KT items for participants.

### Step 3
We developed a *quantitative survey* to assess the psychometric properties of the draft tool.

#### Tool design
The draft tool from step 2 was converted to a questionnaire using Microsoft Excel. Questions were grouped by demographics, the five CFIR domains, and barriers and facilitators of KT.

Demographic questions included country, gender, age, past experience with KT, and professional focus. Items designed to assess readiness were developed on a 5-point Likert scale ('strongly disagree to strongly agree'), following the design of the reviewed validated readiness tools. The final questions asked participants to select the top 3 barriers and facilitators of KT activities from a list generated based on steps 1 and 2. Negatively worded questions were reversed in numeric value, so the number 5 consistently reflected positive attitudes.

We followed the guidelines outlined by Choi and Pak[30] to avoid questionnaire biases. This included refraining from irrelevant redundancies, incorporating largely positively worded items, and using priority references and activities to avoid ambiguity. Open-ended questions were minimised to reduce response fatigue and the format was designed to cluster Likert-scale questions, reducing the length of the questionnaire.

#### Survey population and procedure
The study population for the survey included faculty, staff, and leadership at each of the six academic STRIPE

institutions in addition to members of the Translating Evidence to Action Working Group (TWG) within the Health Systems Global consortium who currently work for academic institutions based in LMICs. The TWG is a multidisciplinary group that comprised approximately 220 researchers, decision-makers and implementers with a focus on the translation of health systems evidence into action (ie, KT) and supporting mechanisms to share best practices globally. STRIPE principal investigators (PI) at each institution identified faculty, staff, and leadership who had been involved in KT at their institution, among whom the survey was then distributed. The eligibility criteria for participating in the survey included the following: (1) must be an active member (eg, staff, faculty, leadership) of an academic institution within the STRIPE consortium or TWG and (2) the institution must be based in an LMIC. A total of 200 eligible participants were identified via the STRIPE consortium and were invited to participate in the survey via Qualtrics, an online survey software, with options to respond in English or French. TWG does not track demographics on its participants, so we do not know, of the 220 members, how many were eligible to complete the survey.

The survey questions were the same for all participants. However, the health priority reference points and list of KT activities identified for each institution in step 2 were used to illustrate the survey questions, and these differed from one academic institution to another. Members of the TWG, all highly familiar with KT, were not provided health priority references but were asked which health issues their institutions prioritised. The links to the Qualtrics surveys were circulated via email through STRIPE PIs, members of their leadership, or through local research assistants. Members of the thematic working group were contacted via email using the group's listserv. Survey data were collected from 6 February 2020 to 25 March 2020.

Responses were monitored for completeness (missing values) and correctness (invalid responses, outliers, and extreme observations). In-country liaisons at each STRIPE institution were asked to follow up with potential respondents who may have started the survey but not completed it.

### Analysis
We conducted exploratory factor analysis (EFA) to identify underlying dimensions of institutional capacity and readiness to conduct KT activities among LMIC institutions. The process was iterative and exploratory, where each observed variable was considered as being a potential measure of factors, and the goal being to determine relationships between observed variables and factors that are strongest. EFA encompasses completing a series of sequential steps that involves evaluation of various options and making a decision at each step. These include assessing sample size adequacy, requisite correlation between variables, appropriate number of factors to include in the model, as well as selection of the factor rotation method.

We began with a sample size (N) to items (p) ratio of approximately 1.5:1. Guiding norms on an adequate sample size for conducting EFA state a desired ratio of at least 5:1,[31] while others agree smaller sample sizes can be justified when higher correlation coefficients are present.[32] Adequacy of sample size was assessed via the Kaiser-Meyer-Olkin (KMO) test. The KMO statistic indicates the proportion of variance in the variables that might be caused by underlying factors. A score above 0.5 is generally regarded as acceptable. We proceeded with a Pearson bivariate correlation of all items, inspecting for markedly high values, which indicate multicollinearity. Ultimately for pairs with correlation coefficients greater than 0.5, one item from each pair was removed based on a qualitative analysis of the variables. Factorability was confirmed by Bartlett's test of sphericity, which tests the null hypothesis that the original correlation matrix is an identity matrix, indicating that variables are unrelated and therefore unsuitable for structure detection. A significant value (<0.05) is requisite. Multicollinearity was explored via the determinant of the matrix, which should be greater than 0.00001.

The survey questions then underwent factor analysis (FA) with an oblique rotation (oblimin). We chose FA over principal component analysis, which assumes the total variance of the variables can be accounted for by means of its components (or factors), and hence that there is no error variance. Alternatively, FA assumes that the variables do not account for 100% of the variance, allowing it to be more flexible. Subsequently, a factor rotation allowed us to obtain a simpler, more interpretable solution for the FA. The choice of rotation depends on whether there is good theoretical reason to suppose the factors should be related or independent of each other. An oblique rotation allows a degree of correlation between the factors. In assessing the number of factors to retain, we considered multiple criteria. Factors with eigenvalues less than 1 were excluded, based on Guttman-Kaiser's rule. Using a scree plot, all factors after the elbow were excluded. Only items that loaded highly (>0.5) and uniquely on each factor were retained. Factors with fewer than three variables loading on to it were also excluded. The proportion of the total variance explained by the retained factors was examined to be above the recommended minimum of 50%. Finally, Cronbach's alpha coefficient was calculated to test for scale reliability and internal consistency within and between factors. The norm is to accept values 0.7 or higher.

Data analysis was conducted in Stata V.13.[33]

## RESULTS
### Literature review
Review of the literature revealed 26 organisational readiness tools related to health institutions (including hospitals, clinics, research institutes, etc) and 30 tools developed specifically to assess organisational readiness for implementing global health interventions; only one

was designed specifically for KT. Both sets of tools had been collated and analysed in systematic reviews.[23 26] Common constructs across these tools included availability of resources, individual attributes and motivation, and organisational climate.

## Stakeholder interviews

Eighteen stakeholders, both internal and external to the six STRIPE academic institution, were interviewed. Three cross-cutting themes emerged as relevant for readiness to conduct KT in LMICs. These included (1) the complexity of the policy process and necessity of 'soft-skills' (ie, communication, self-awareness, relationship maintenance), (2) misalignment between institutional missions and incentives, and (3) the role of internal and external networks. The results from the consultation process are described in more detail elsewhere.[9] The constructs identified from the literature review and the cross-cutting themes from the consultation process were organised into a quantitative tool with 5 domains and 76 items, with new items developed for the themes. A total of nine additional questions on demographics, facilitators, and barriers to doing KT were added to translate the tool into a survey questionnaire (online supplemental appendix 1).

## Survey

We received 158 responses to the survey across the 6 STRIPE countries and TWG; 148 (74%) of the eligible participants from the STRIPE countries responded to the survey. There were 47 respondents who completed 9% or less of the survey, which were subsequently dropped from the analysis. These respondents did not cluster by country, age or gender and appeared random. A total of 111 responses were included in the final analysis. According to Arrindell and van der Ende,[32] a sample size (N) to items (p) ratio, that is, N:p ratio, of 3:1 is adequate for demonstrating a stable factor structure with an alpha level of 0.05. Hence, our sample size of 111 is adequate for demonstrating the validity and reliability of a tool with at least 37 items.

There were 27 respondents from Bangladesh, 19 from Indonesia, 16 from India, 16 from Nigeria, 12 from DRC, 11 from Ethiopia, and 10 from the TWG representing other LMICs. Of the total 111 respondents, 53% (n=59) were male and 43% (n=48) were female. Majority of the respondents (57%, n=64) were 30–49 years of age and most (59%, n=66) indicated they had experience in conducting KT. Participants were asked to indicate their current professional focus/foci. Most respondents were engaged in research (83%, n=93) and teaching (62%, n=69). Other common foci included project coordination (36%, n=41), leadership (15%, n=17) and management (16%, n=18).

The most conducted KT activities included 'taught a course on communication, advocacy, stakeholder engagement or KT' (98%, n=109), 'conducted a stakeholder meeting' (62%, n=69) and 'given a presentation at a scientific conference' (60%, n=67). Individuals who indicated

having experience with KT activities, compared with those with no experience, were significantly more likely to have written a policy brief (p=0.0059), conducted a stakeholder meeting (p=0.0364), engaged with policy makers to set priorities (p=0.0129), and to have given a presentation at a scientific conference (p=0.0011). Two KT activities also varied significantly by country: 'authored or co-authored an article in a peer-review journal' (p=0.0001) and 'given a presentation at a scientific conference' (p=0.0002). Additional descriptive statistics are presented in table 2.

## Exploratory factor analysis

We ran an EFA on the complete data set (version 0), which included 76 items. This approach yielded 22 factors; both the KMO measure and Barlett's test yielded no value. Constructs covered in these factors included individual motivation, organisational climate, organisational culture, internal resources, individual knowledge and skills, internal and external networks, funding sources, prioritisation, and shared ethos for change (change valence). Many factors overlapped, each addressing similar or related constructs. The correlation matrix was reviewed for highly correlated items and those with correlations above 0.5 were dropped; 17 items were removed in this process. Highly correlated items included 'Q1: I am confident that I can conduct KT activities', 'Q3: I feel personally motivated to do KT' and 'Q10: I have the skills to conduct KT'. Q10 was also heavily correlated with 'Q9: I know how to do KT' and 'Q11: I have experience conducting KT'. Wherever possible, items were kept that did not correlate heavily with other items. These items were also reviewed to ensure they captured the same or similar information.

For the remaining 59 items (version 1.0) we repeated the EFA, followed by an oblique rotation, producing as simple a structure as possible while permitting correlations among factors. This yielded 17 factors, with a KMO measure of 0.4 and a significant value for Bartlett's test (p=0.000). The items, their median and IQR for version 1.0 can be found in online supplemental appendix 2. Individual motivation, networks, prioritisation, organisational climate and resources were still captured by these factors. New constructs emerged in this model, including institutional peer pressure, the process of conducting KT, and perceived value of KT.

All cross-loading items (ie, items loading on more than one factor) or with a loading less than 0.5 were then dropped, resulting in the removal of 26 additional items (version 2.0). Table 3 shows each item included in version 2.0, the item's median, and IQR. Some dropped items addressed individual motivation (eg, 'Q13: I am passionate about conducting KT' and 'KT activities have a positive impact on the health of communities') and institutional climate (eg, 'Q32: My institution provides opportunities for professional development in KT' and 'Q59: I have at least one mentor who conducts KT with the ministry of health'). We reran the FA with 31 items, followed by oblique rotation, and identified 7 factors. The

**Table 2** Characteristics of respondents

| Variable | Bangladesh n=27 (%) | DRC n=12 (%) | Ethiopia n=11 (%) | India n=16 (%) | Indonesia n=19 (%) | Nigeria n=16 (%) | Others n=10 (%) | Total n=111 (%) |
|---|---|---|---|---|---|---|---|---|
| Age (years) | | | | | | | | |
| 18–29 | 13 (48.15) | 0 (0.00) | 0 (0.00) | 0 (0.00) | 6 (31.58) | 0 (0.00) | 0 (0.00) | 19 (17.12) |
| 30–49 | 12 (44.44) | 6 (50.00) | 8 (72.73) | 7 (43.75) | 11 (57.89) | 12 (75.00) | 8 (80.00) | 64 (57.66) |
| 50–69 | 2 (7.41) | 5 (41.67) | 3 (27.27) | 7 (43.75) | 0 (0.00) | 3 (18.75) | 2 (20.00) | 22 (19.82) |
| ≥70 | 0 (0.00) | 1 (8.33) | 0 (0.00) | 0 (0.00) | 0 (0.00) | 0 (0.00) | 0 (0.00) | 1 (0.90) |
| Gender | | | | | | | | |
| Male | 16 (59.26) | 6 (50.00) | 8 (72.73) | 10 (62.50) | 5 (26.32) | 9 (56.25) | 5 (50.00) | 59 (53.15) |
| Female | 9 (33.33) | 5 (41.67) | 3 (27.27) | 6 (37.50) | 14 (73.68) | 7 (43.75) | 4 (40.00) | 48 (43.24) |
| Professional focus | | | | | | | | |
| Research | 22 (81.48) | 12 (100.00) | 10 (90.91) | 14 (87.50) | 14 (73.68) | 13 (81.25) | 8 (80.00) | 93 (83.78) |
| Administration | 1 (3.70) | 1 (8.33) | 0 (0.00) | 1 (6.25) | 1 (5.26) | 5 (31.25) | 0 (0.00) | 9 (8.11) |
| Leadership | 1 (3.70) | 1 (8.33) | 0 (0.00) | 4 (25.00) | 3 (15.79) | 6 (37.50) | 2 (20.00) | 17 (15.32) |
| Project coordination | 6 (22.22) | 6 (50.00) | 1 (9.09) | 12 (75.00) | 8 (42.11) | 5 (31.25) | 3 (30.00) | 41 (36.94) |
| Communications | 1 (3.70) | 0 (0.00) | 0 (0.00) | 3 (18.75) | 1 (5.26) | 1 (6.25) | 2 (20.00) | 8 (7.21) |
| External affairs | 0 (0.00) | 2 (16.67) | 0 (0.00) | 0 (0.00) | 1 (5.26) | 0 (0.00) | 0 (0.00) | 3 (2.70) |
| Development | 0 (0.00) | 0 (0.00) | 0 (0.00) | 3 (18.75) | 1 (5.26) | 1 (6.25) | 1 (10.00) | 6 (5.41) |
| Management | 4 (14.81) | 1 (8.33) | 0 (0.00) | 6 (37.50) | 3 (15.79) | 3 (18.75) | 1 (10.00) | 18 (16.22) |
| Teaching | 7 (25.93) | 12 (100.00) | 10 (90.91) | 14 (87.50) | 4 (21.05) | 15 (93.75) | 7 (70.00) | 69 (62.16) |
| Finance | 0 (0.00) | 0 (0.00) | 0 (0.00) | 0 (0.00) | 0 (0.00) | 0 (0.00) | 0 (0.00) | 0 (0.00) |
| IT | 0 (0.00) | 0 (0.00) | 0 (0.00) | 1 (6.25) | 0 (0.00) | 0 (0.00) | 0 (0.00) | 1 (0.90) |
| Regulatory services | 2 (7.41) | 1 (8.33) | 0 (0.00) | 1 (6.25) | 0 (0.00) | 0 (0.00) | 1 (10.00) | 5 (4.50) |
| KT experience | | | | | | | | |
| Yes | 14 (51.85) | 5 (41.67) | 7 (63.64) | 14 (87.50) | 12 (63.16) | 11 (68.75) | 3 (30.00) | 66 (59.46) |
| No | 9 (33.33) | 1 (8.33) | 1 (9.09) | 1 (6.25) | 2 (10.53) | 4 (25.00) | 3 (30.00) | 21 (18.92) |
| Unsure | 4 (14.81) | 6 (50.00) | 3 (27.27) | 1 (6.25) | 4 (21.05) | 1 (6.25) | 3 (30.00) | 22 (19.82) |
| KT activities | | | | | | | | |
| Written a policy brief | 10 (37.04) | 3 (25.000 | 2 (18.18) | 8 (50.00) | 11 (57.89) | 7 (43.75) | 3 (30.00) | 44 (39.64) |
| Written an evidence summary | 5 (18.52) | 4 (33.33) | 4 (36.36) | 7 (43.75) | 7 (36.84) | 5 (31.25) | 3 (30.00) | 35 (31.53) |
| Conducted a stakeholder meeting | 12 (44.44) | 7 (58.33) | 5 (45.45) | 12 (75.00) | 16 (84.21) | 10 (62.50) | 7 (70.00) | 69 (62.16) |
| Conducted a policy dialogue | 3 (11.11) | 1 (8.33) | 0 (0.00) | 6 (37.50) | 5 (26.32) | 2 (12.50) | 2 (20.00) | 19 (17.12) |
| Engaged with an advocacy campaign | 5 (18.52) | 3 (25.00) | 1 (0.09) | 7 (43.75) | 6 (31.58) | 7 (43.75) | 2 (20.00) | 31 (27.93) |
| Engaged with policy makers to set priorities | 3 (11.11) | 4 (33.33) | 3 (27.27) | 6 (37.50) | 9 (47.37) | 5 (31.25) | 4 (40.00) | 34 (30.63) |
| Developed a video for a policy maker | 5 (18.52) | 1 (8.33) | 0 (0.00) | 4 (25.00) | 4 (21.05) | 2 (12.50) | 1 (10.00) | 17 (15.32) |
| Engaged with the media | 5 (18.52) | 3 (25.00) | 2 (18.18) | 8 (50.00) | 5 (26.32) | 8 (50.00) | 5 (50.00) | 36 (32.43) |

Continued

**Table 2** Continued

| Variable | Bangladesh n=27 (%) | DRC n=12 (%) | Ethiopia n=11 (%) | India n=16 (%) | Indonesia n=19 (%) | Nigeria n=16 (%) | Others n=10 (%) | Total n=111 (%) |
|---|---|---|---|---|---|---|---|---|
| Used a KT platform | 4 (14.81) | 4 (33.33) | 2 (18.18) | 7 (43.75) | 4 (21.05) | 0 (0.00) | 3 (30.00) | 24 (21.62) |
| Authored or coauthored | 8 (29.63) | 9 (75.00) | 7 (63.64) | 11 (68.75) | 7 (36.84) | 16 (100.00) | 9 (90.00) | 67 (60.36) |
| Conducted a systematic or rapid review | 6 (22.22) | 1 (8.33) | 6 (54.55) | 5 (31.25) | 2 (10.53) | 4 (25.00) | 4 (40.00) | 28 (25.23) |
| Taught a course on communication, advocacy, stakeholder engagement or KT | 27 (100.00) | 12 (100.00) | 11 (100.00) | 14 (87.50) | 19 (100.00) | 16 (100.00) | 10 (100.00) | 109 (98.20) |
| Worked with a journalist to disseminate information | 2 (7.41) | 3 (25.00) | 0 (0.00) | 6 (37.50) | 3 (15.79) | 4 (25.00) | 2 (20.00) | 20 (18.02) |
| Given a presentation at a scientific conference | 8 (29.63) | 7 (58.33) | 6 (54.55) | 14 (87.50) | 9 (47.37) | 15 (93.75) | 8 (80.00) | 67 (60.36) |

Data were missing for some variables, therefore numbers do not always add to the total.
DRC, Democratic Republic of the Congo; IT, information technology; KT, knowledge translation.

model produced a KMO of 0.52 and a significant value for Barlett's test (p=0.000). In this model one item ('Q38: If I want to conduct a KT activity, I know where to find people in my institution who can help') loaded on two factors and was thus removed; in the resulting model, another item loaded to two factors and the item was dropped. A final set of 31 items were retained.

We reran the FA with the 31 retained items to identify a five-factor model. Items were selected based on the strength of the factor loading, uniqueness of the factors, the resulting scree plot, and cross-loading criteria, and we selected 23 items for the 5 factors, with N:p ratio of 5:1. Figure 1 displays the eigenvalue plot of the final factor model. For the final five-factor model, the average communality (also known as uniqueness) of selected items was 0.61. The items (p) per factor (r) ratio (p:r ratio) in the final model presented in table 3 is 23:5 with N=111. According to MacCallum et al,[31] sample sizes of N=100 and N=200 are needed to estimate stable factor structure with 95% convergence for p:r ratio of 20:3 and 10:3, respectively, if the average communality is low (less than 0.4). If the communality is high (>0.4), as found in our study, N=60 is adequate for p:r ratio of 10:3 or 20:3 and will estimate the factor structure with over 99% convergence.

The five final factors that emerged from the analysis were named (1) institutional climate, (2) organisation change efficacy, (3) prioritisation and cosmopolitanism, (4) self-efficacy, and (5) financial resources, based on their item characteristics and the underlying theories for those items.[9] Factor 1 contains six items and was labelled 'institutional climate' because each item described aspects of their institution, colleagues, and leadership. Factor 2 contains five items and was labelled 'organisation change efficacy' to capture organisational members' shared beliefs in their joint abilities. Factor 3, 'prioritisation and cosmopolitanism', which also comprises five items, relates to internal and external institutional networks and priorities. Factor 4, which comprises four items, captures individual influencers of 'self-efficacy', including knowledge, skills, and time. Factor 5, 'financial resources', contains items related to internal and external budgets for KT activities.

Based on data collected during the stakeholder consultation process, these factors demonstrate face and content validity. That is, they appear to measure factors relevant to KT in these settings and represent the complex facets of the constructs.

These five factors combined accounted for 69% of the total variance. The factor loadings, which ranged from 0.40 to 0.77, are presented in table 4; the intercorrelations between the factors ranged from 0.04 to 0.31. The final model observed a KMO measure of 0.554 and the Bartlett's test of sphericity was significant ($\chi^2$ (465)=771.570, p<0.000). Cronbach's alpha coefficients were 0.78, 0.73, 0.62, 0.68 and 0.52 respectively. Factors 1 and 2 report an alpha above 0.7, which is traditionally acceptable.[34 35] Although there are no firm bounds on Cronbach's alpha,

**Table 3** Survey questions (version 2) with median (IQR)

| Survey questions | Median (IQR) |
|---|---|
| Q26. In general in my institution when there is agreement that KT needs to happen we have the necessary support in terms of training. | 3 (2–4) |
| Q30. Senior leadership/clinical management in my institution rewards innovation and creativity to improve KT. | 2 (1–3) |
| Q31. Financial incentives are available for me to conduct KT (eg, bonus salary). | 3 (2–5) |
| Q37. My institution provides trainings on knowledge translation activities. | 3 (1–3) |
| Q38. If I want to conduct a KT activity, I know where to find people in my institution who can help. | 4 (3–4) |
| Q39. Senior members/leadership of my institution provide me with connections to conduct KT. | 2 (2–3) |
| Q40. People within my institution talk about their KT activities with each other. | 2 (1–3) |
| Q56. Other faculty and staff members are available to collaborate on KT activities. | 2 (1–3) |
| Q2. People at my institution are confident they can conduct KT activities. | 2 (2–4) |
| Q6. I feel personally motivated to do KT because I will be punished by my institution if I do not. | 5 (3–5) |
| Q7. Others in my institution feel motivated to do KT. | 2 (1–2) |
| Q44. If my institution does not conduct KT with the ministry, another college or university in my country will. | 4 (3–4) |
| Q45. The funding organisations that support my research require KT activities. | 4 (3–4) |
| Q46. The ministry relies on my institution more than other institutions to conduct KT. | 2 (1–3) |
| Q47. Other institutions do more KT than my institution. | 3 (2–3) |
| Q51. Members of my government understand the importance of scientific data for making decisions about health. | 2 (2–4) |
| Q52. Members of my government want to work with my institution to improve health. | 4 (3–4) |
| Q54. Ministry members in my country prefer policy briefs to other forms of KT activities. | 4 (3–4) |
| Q69. When conducting KT activities, it is important to engage a wide range of stakeholders. | 4 (2–4) |
| Q41. My institution includes KT in its strategic plan, mission or vision. | 2 (1–4) |
| Q49. Ministry members and politicians in my country make health decisions without scientific consideration. | 2 (2–4) |
| Q60. Senior members/leadership of my institution use their networks to help others conduct KT. | 2 (2–4) |
| Q66. I spend a lot of time planning my KT activities. | 2 (1–3) |
| Q71. When I conduct KT activities, they address current priorities of the ministry. | 2 (1–3) |
| Q76. I am aware of donors that fund KT activities. | 3 (2–4) |
| Q11. I have experience conducting KT. | 4 (3–4) |
| Q12. I have received training to conduct KT activities. | 3 (2–5) |
| Q15. I have time to dedicate to KT in addition to my other tasks. | 2 (2–4) |
| Q20. I know how to translate my data and key findings for policy makers. | 2 (2–3) |
| Q55. KT teams at my institution have clearly defined roles and responsibilities. | 2 (1–3) |
| Q53. Most projects I am involved with have budgeted for communications and advocacy activities. | 2 (1–3.5) |
| Q63. Conducting KT activities is more of an art than a science. | 2 (2–4) |
| Q75. Financial resources are available at the Ministry of Health to support the cost of KT. | 2 (1–3) |

KT, knowledge translation.

factors 3–5 report a lower alpha, which can indicate a need for cautious interpretation. However, given the theoretical underpinnings for each of these factors related both to concepts of individual and institutional readiness, and the small number of items loading to each factor, we concluded the Cronbach's alpha values are still helpful.[36]

## DISCUSSION
Assessing readiness of individuals and institutions in LMICs is different from that in HICs. While some approaches, barriers, and facilitators are shared, low-resource settings have unique contexts that influence KT processes, the individuals who conduct KT, and the organisations in which they operate. We sought to develop a robust organisational readiness tool designed to reflect these contextual factors specifically for KT activities. We know that while LMICs are diverse, certain barriers and determinants of institutional capacity to conduct KT are generalisable across diverse LMIC contexts,[9] and evaluating readiness of academic institutions to conduct KT along these barriers and factors may provide a starting point for enacting KT practices in specific LMICs. However, like most KT and implementation research endeavours, such readiness assessments

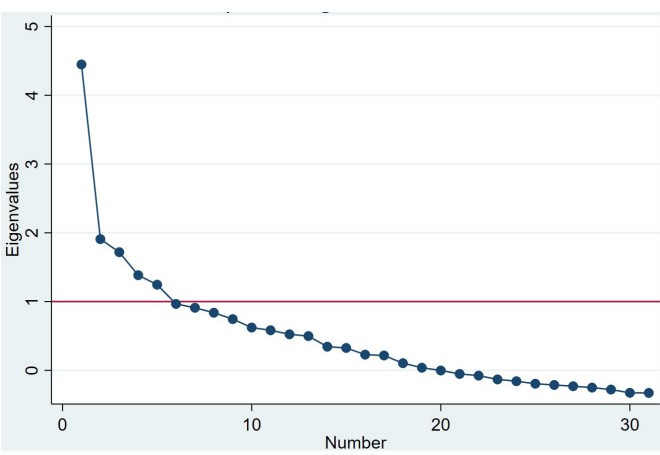

**Figure 1** Final factor model scree plot.

or any relevant tools will have to be further adapted to specific country contexts. The aim of this study is to provide those initial constructs that may be adaptable to different LMIC academic institutions for assessing their readiness to conduct KT activities.

Five factors emerged as relevant for readiness to conduct KT in LMICs: institutional climate, organisation change efficacy, prioritisation and cosmopolitanism, self-efficacy, and financial resources.

### Institutional climate

Institutional climate conceptualises how individuals perceive and describe their work setting.[37] This can include shared perceptions of what is rewarded within an institution and what is expected of people in their roles, in other words organisational members' shared beliefs and values. The items included in this factor highlight different components of climate, including rewards for innovation and creativity, financial incentives to conduct KT, and provision of trainings. Other items describe concepts of shared values through colleague collaboration, both in practice and availability to collaborate. Some interview participants described this as an enabling environment which could include access to infrastructure but also to networks:

> An enabling environment is where you have everything you need. In the ministry you have the internet, these laptops, these digital tools, the supportive director and supportive leadership who has recognized the importance of knowledge translation and is using it to influence change. —Nigeria, external 1

> I think having mentors that consider knowledge translation important and that prioritize knowledge translation would be a great motivating factor. Mentors, as well as senior colleagues…but just seeing other people doing it, and seeing how they do it, and potentially how it can be rewarding, I think, is helpful and encouraging individual researchers to also conduct knowledge translation. —Nigeria, internal 2

### Organisation change efficacy

Weiner[16] first described this concept as 'organizational members' shared beliefs in their collective capabilities to organize and execute the courses of action involved in change implementation'. In 2014, Shea *et al*[19] assessed the psychometric properties of scales developed to measure change efficacy and other facets of organisational readiness. The change efficacy items focused on confidence in abilities to manage change processes, coordinate tasks, maintain momentum and get investment. Items mapped to this factor from our scale similarly describe confidence and motivation (ie, investment) and further contextualise this for KT. For example, items Q46, Q51 and Q69 account for the perceived role of external stakeholders, including the MoH and members of the government, and their views on data and reliance on the institution, to conduct KT. Organisation members may commit to change because they value it, have little choice, or feel obliged[16]—it makes sense that the views and actions of external stakeholders could influence those values, feelings, and obligations, particularly for health-related institutions.

### Prioritisation and cosmopolitanism

Public health institutes are increasingly called on to align and adapt their activities to the health priorities of the country. Major international funders including the US Centers for Disease Control and Prevention have recently released calls for applications that address this issue of priority alignment between country governments and in-country public health organisations. When applied to academic organisations, this tool highlighted the role of institutional strategies, missions and visions, and the importance of conducting KT activities that address national priorities. Integral to this concept is also the extent to which organisational networks can be leveraged to conduct KT. Bloland *et al*[38] argue that an important priority for public health institutions is to collaborate with MoHs to improve their abilities to not only accumulate data, but also manage that knowledge and translate it into actionable policies. This network component is fundamental to prioritisation.

### Self-efficacy

Readiness has often referred to an individual psychological state of motivation and plays an important role in many theories of behaviour change, including the health belief model (eg, self-efficacy),[39] Prochaska's stages of change model (eg, determination)[40] and the social cognitive theory (eg, capability and self-efficacy).[41] It is unsurprising that individual-level factors such as knowledge, training, and roles emerged as an important factor in this analysis. One unique item assessing time to dedicate to KT, included in this factor, was developed through data that emerged from the qualitative interviews. Participants reflected on competing priorities for their time and the need for protected and financially supported time to conduct KT.

**Table 4** Factor loadings, eigenvalues and uniqueness

| Survey questions | 1 | 2 | 3 | 4 | 5 | Eigenvalue | Uniqueness |
|---|---|---|---|---|---|---|---|
| Q96_8. Senior leadership/clinical management in my institution rewards innovation and creativity to improve KT. | 0.5919 | | | | | 4.44856 | 0.6234 |
| Q96_9. Financial incentives are available for me to conduct KT (eg, bonus salary). | 0.4766 | | | | | | 0.4502 |
| Q96_15. My institution provides trainings on knowledge translation activities. | 0.5384 | | | | | | 0.6646 |
| Q96_17. Senior members/leadership of my institution provide me with connections to conduct KT. | 0.7741 | | | | | | 0.5035 |
| Q96_18. People within my institution talk about their KT activities with each other. | 0.6051 | | | | | | 0.6781 |
| Q98_2. Other faculty and staff members are available to collaborate on KT activities. | 0.4296 | | | | | | 0.5118 |
| Q11_2. People at my institution are confident they can conduct KT activities. | | 0.5776 | | | | 1.90881 | 0.6893 |
| Q11_7. Others in my institution feel motivated to do KT. | | 0.7265 | | | | | 0.6405 |
| Q97_5. The ministry relies on my institution more than other institutions to conduct KT. | | 0.4051 | | | | | 0.3678 |
| Q97_10. Members of my government understand the importance of scientific data for making decisions about health. | | 0.4237 | | | | | 0.5058 |
| Q98_15. When conducting KT activities, it is important to engage a wide range of stakeholders. | | 0.4826 | | | | | 0.543 |
| Q46. My institution includes KT in its strategic plan, mission or vision. | | | 0.6441 | | | 1.71858 | 0.7296 |
| Q97_8. Ministry members and politicians in my country make health decisions without scientific consideration. | | | 0.4222 | | | | 0.6502 |
| Q98_6. Senior members/leadership of my institution use their networks to help others conduct KT. | | | 0.4218 | | | | 0.7574 |
| Q98_12. I spend a lot of time planning my KT activities. | | | 0.4568 | | | | 0.6711 |
| Q99_2. When I conduct KT activities, they address current priorities of the ministry. | | | 0.5799 | | | | 0.5658 |
| Q94_5. I have received training to conduct KT activities. | | | | 0.4011 | | 1.38363 | 0.6807 |
| Q95_2. I have time to dedicate to KT in addition to my other tasks. | | | | 0.6935 | | | 0.6691 |

Continued

**Table 4** Continued

| Survey questions | 1 | 2 | 3 | 4 | 5 | Eigenvalue | Uniqueness |
|---|---|---|---|---|---|---|---|
| Q95_7. I know how to translate my data and key findings for policy makers. | | | | 0.4816 | | | 0.4893 |
| Q98_1. KT teams at my institution have clearly defined roles and responsibilities. | | | | 0.432 | | | 0.6713 |
| Q97_12. Most projects I am involved with have budgeted for communications and advocacy activities. | | | | | 0.434 | 1.24604 | 0.6061 |
| Q98_9. Conducting KT activities is more of an art than a science. | | | | | 0.6443 | | 0.5737 |
| Q99_6. Financial resources are available at the Ministry of Health to support the cost of KT. | | | | | 0.5578 | | 0.6743 |

Values <0.4 are suppressed.
Factor 1: 'institutional climate'; factor 2: 'organisation change efficacy'; factor 3: 'prioritisation and cosmopolitanism'; factor 4: 'self-efficacy'; factor 5: 'financial resources'.
KT, knowledge translation.

We are researchers, and we are trained to do research and the research also takes up a lot of our time and energy. And knowledge translation itself takes a lot of time and energy and a completely different skill set; it's not a research skill. —Indonesia, internal 1

### Financial resources

KT activities, like most research and programmatic work, require the availability of financial resources. Items loading to this factor indicated that this is true not just for the academic institution (eg, project budgets), but also at the MoH. KT models for evidence sharing frequently describe knowledge-push (knowledge supplied by researchers), demand-pull (demand for knowledge from policy makers) and interactive approaches.[42–44] These underscore that KT is a dynamic process, requiring time, input, and resources from knowledge generators, translators, and users. Given the nuances of this process it is interesting but unsurprising that the item 'Conducting KT activities is more of an art than a science' loaded to this factor with a focus on resources.

Components of each of these five factors have been considered by existing organisational readiness tools, although none has been combined in this way or used for KT specifically.[23] This FA builds on existing measurement tools and further demonstrates the dynamic nature of KT and underscores important contextual considerations for LMIC institutions. This includes the role of internal collaborations (institutional climate) and external networks (prioritisation and cosmopolitanism), which often rely on leadership and senior institutional members. Literature has also noted the important role that funding organisations play in supporting KT in LMICs, both through a prioritisation and a financial lens.[45]

The final five-factor model captures many of the concepts represented in the original model but condenses constructs related to the individual (eg, motivation and knowledge became self-efficacy) and the organisation (eg, different constructs of climate merged). Prioritisation and cosmopolitanism, two constructs that capture internal and external networks, are consolidated in the final model, demonstrating the relationship between priority setting and networks. The outer setting and outer needs and resources, present across factors in the original model, are represented by organisation change efficacy and financial resources, highlighting how external factors can influence an organisation's capacity to conduct KT. Other readiness domains not captured in the final model may not have been as relevant for the population studied across these settings but may still have relevance for other types of academic institutions in different LMICs.

This is the first organisational readiness tool designed for KT activities in LMICs. The model can be used as a starting point and adapted by LMIC institutions to assess their readiness for KT, understand implementation challenges of KT initiatives, explore facilitators and barriers, and provide quantitative measurements for these institutions. The model can also be used to explore determinants of KT in these settings to inform the development of tailored strategies to improve institutional and individual capacity to conduct KT.

### Strengths and limitations

The development of this tool was rooted in organisational change theory, building on validated tools. We conducted the research across multiple contexts in both Africa and Asia to capture constructs generalisable to institutions based in LMICs. A significant limitation of this research

is the small sample size. Most well-accepted guidelines for sample size to conduct EFA recommend a ratio of 5:10 subjects per item. However, studies have argued smaller sample sizes may be justified when higher correlation coefficients are observed.[46] The original tool was lengthy, which could have resulted in response fatigue, leading to inaccurate data or incomplete responses. This may have been further exacerbated by selection bias if participants whose responses were dropped were similar in some way. Finally, we noted that participants from DRC and Ethiopia were possible outliers, skewing older (and therefore likely more experienced with KT) and more male than respondents from other countries while also having fewer responses than many of the other settings.

## CONCLUSION

KT is perceived as valuable for bridging the 'know-do' gap, bringing evidence-based interventions into policies and practice. The five factors that emerged from this research as relevant to readiness to conduct KT highlight unique contextual influencers and opportunities for capacity strengthening in LMICs. The organisational focus of these factors further points to a need for capacity building that includes but goes beyond individual training. Future research will be conducted to further understand the influencers of these readiness factors and systematically develop capacity building strategies for academic institutions in LMICs to conduct KT.

**Acknowledgements** The authors would like to acknowledge the team members of the academic institutions included under the STRIPE academic consortium: Dr Piyusha Majumdar, Dr SD Gupta and Dr DK Mangal of the IIHMR University, India, Dr Riris Andono Ahmad, Dr Yodi Mahendradhata and Ms Utsamani Cintyamena of Universitas Gadjah Mada, Indonesia, Dr Assefa Seme of Addis Ababa University, Ethiopia, Dr Eme Owoaje and Dr Oluwaseun Akinyemi of the University of Ibadan, Nigeria, Dr Malabika Sarker of BRAC University, Bangladesh, and Dr Patrick Kayembe of the Kinshasa School of Public Health for their support in data collection. The authors would also like to thank the study participants for their time and contributions to the study.

**Contributors** AK conceived the study and paper with guidance from OA. AK conducted the literature review, stakeholder engagement, tool development and tool distribution and authored all drafts of the manuscript. AR supported the data analysis and contributed to authorship of the initial draft of the paper. OA provided significant oversight throughout the conceptualisation, analysis and write-up process, and edited each draft. All authors read and approved the final manuscript.

**Funding** The parent project to this research, STRIPE, is funded by the Bill and Melinda Gates Foundation (OPP1178578).

**Competing interests** None declared.

**Patient consent for publication** Not required.

**Ethics approval** This research was declared 'Non-Human Subjects' by the Johns Hopkins Bloomberg School of Public Health Institutional Review Board on 8 October 2019. All participants were read a short statement on consent and provided a verbal agreement prior to the interview.

**Provenance and peer review** Not commissioned; externally peer reviewed.

**Data availability statement** Data are available upon reasonable request. The datasets used and/or analyzed during the current study are available from the corresponding author on reasonable request.

**ORCID iD**
Anna Kalbarczyk http://orcid.org/0000-0002-6143-8634

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
