## [Reviewer comments · BMJ Open]

ARTICLE DETAILS

TITLE (PROVISIONAL)	A mixed methods study to develop a tool to assess institutional readiness to conduct knowledge translation activities in low-and middle-income countries
AUTHORS	Kalbarczyk, Anna; Rao, Aditi; Alonge, Olakunle

VERSION 1 – REVIEW

REVIEWER	Kele, Pierre Abomo LSTM, International Public Health
REVIEW RETURNED	09-May-2021

GENERAL COMMENTS	Building a tool to measure the readiness of academic institutions in KT in LMICs is a daunting task. The concept of LMICs is itself too heterogeneous a category to result in acceptable categories and generalizations. Whether it is the health system or the education system, it is difficult to include the so-called LMIC countries under the same category. Just to take the example of sub-Saharan Africa, you mentioned South Africa in the model. But South Africa is a context might be said to be looks closer to that of HICs than LMICs. Likewise, other factors such as the Anglophone, Francophone or Lusophone context are important variables that will influence the design of your tool, each time bringing fundamental differences in readiness to KT. Without finely analyzing the context of application of the tool, without adopting an à la carte approach, your tool will have difficulty grasping the full complexity of LMICs. Perhaps a prior analysis of this context and a good categorization of LMICs would lead to, not one tool, but several adaptable depending on the country.
---

REVIEWER	Oronje, Rose African Institute for Development Policy (AFIDEP)
REVIEW RETURNED	31-May-2021

GENERAL COMMENTS	The paper makes an important contribution to knowledge by producing a tool for assessing LMIC institutions readiness for undertaking KT. This is an important tool that could be instrumental in institutions' efforts to improve KT, and therefore enhance their impact on health policy decisions and health care service provision. Specific comments: 1. Methods section: a) How was "step 2" conducted? This is the step on stakeholder consultations - was this done through a workshop, a series of workshops, or one-on-one interviews? Some where in the Results section you indicate that "18 stakeholders were interviewed" - if
---

	step 2 was indeed interviews, why not just call it "interviews" rather than stakeholder consultations? b) On page 9, you indicate the "this process was also used to confirm health priorities..." referring to step 2. Again, how was the confirming of health priorities done during step 2? Had priorities already been identified and the consultations just commented on these to confirm them? It will be useful to be comprehensive and transparent in explaining how this step 2 was conducted. c) On Step 3 - one of the questions I had when I read this section is "how many of the 200 sampled respondents responded to the survey?" I see that this comes in the Results section and am not that's the right place. d) On Step 3 - I hope other reviewers of this paper assess the quality and relevance of the quantitative analysis done as I don't have the expertise to comment on this analysis 2. Results section: a) I think the content on page 13 fits better in your Methods section than your Results section. b) Results section could benefit from sub-headings 3. Discussion section: a) I am struggling with having the 5 factors presented in this section - why not present these factors in the Results section, and the use the Discussion to present the implications of the tool, and perhaps how different this tool is from tools used in HICs? It seems to me that the content on pages 25, 26 and 27 could well fit in the Results section. b) On page 28, please provide the reference for the statement made in line 5 and 6. Finally, the paper may be benefit from another read to pick out s few typos, e.g. on page 21 line 11.
--	--

VERSION 1 – AUTHOR RESPONSE

Reviewer: 1

Dr. Pierre Abomo Kele, LSTM

Comments to the Author:

Building a tool to measure the readiness of academic institutions in KT in LMICs is a daunting task. The concept of LMICs is itself too heterogeneous a category to result in acceptable categories and generalizations. Whether it is the health system or the education system, it is difficult to include the so-called LMIC countries under the same category. Just to take the example of sub-Saharan Africa, you mentioned South Africa in the model. But South Africa is a context might be said to be looks closer to that of HICs than LMICs. Likewise, other factors such as the Anglophone, Francophone or Lusophone context are important variables that will influence the design of your tool, each time bringing fundamental differences in readiness to KT. Without finely analyzing the context of application of the tool, without adopting an à la carte approach, your tool will have difficulty grasping the full complexity of LMICs. Perhaps a prior analysis of this context and a good categorization of LMICs would lead to, not one tool, but several adaptable depending on the country.

Thank you very much for your helpful comment.

We agree that generalization of measures of Knowledge Translation (KT) readiness across academic institutions in LMIC may be quite challenging – and have made revisions to acknowledge this fact in the manuscript (see page #4 in the introduction section and page #26 in the discussion section).

However, we draw attention to our prior qualitative study conducted across 6 LMICs which identified generalizable facilitators and barriers (and their related determinants) to institutional capacity to conduct KT across diverse LMIC setting – and how that study and this present study (similarly conducted across diverse LMIC settings) might present a starting point for addressing these barriers through assessment of organizational readiness to do KT. We recognize that the five concepts and their related tools (on institutional climate, organization change efficacy, prioritization and cosmopolitanism, self-efficacy, and financial resources) identified for KT readiness assessment may require further contextual adaptation in specific LMIC, but that they provide a starting point for such assessment since they are based on factors that are common across multiple LMICs (see page #26 and 30 in discussion). We hope these revisions provide clarity to the readers and a more accurate reflection of the relevance of this work – and how it can be applied further.

Reviewer: 2

Dr. Rose Oronje, African Institute for Development Policy (AFIDEP), NIMR/HQ/R Comments to the Author:

The paper makes an important contribution to knowledge by producing a tool for assessing LMIC institutions readiness for undertaking KT. This is an important tool that could be instrumental in institutions' efforts to improve KT, and therefore enhance their impact on health policy decisions and health care service provision.

Thank you very much for your helpful comments

Specific comments:

1. Methods section:

a) How was "step 2" conducted? This is the step on stakeholder consultations - was this done through a workshop, a series of workshops, or one-on-one interviews? Some where in the Results section you indicate that "18 stakeholders were interviewed" - if step 2 was indeed interviews, why not just call it "interviews" rather than stakeholder consultations?

We've changed the language to 'stakeholder interviews' to be more clear in this step. Thank you.

b) On page 9, you indicate the "this process was also used to confirm health priorities..." referring to step 2. Again, how was the confirming of health priorities done during step 2? Had priorities already been identified and the consultations just commented on these to confirm them? It will be useful to be comprehensive and transparent in explaining how this step 2 was conducted.

The health priorities were initially identified during the literature review (including gray literature from government and ministries of health in the countries), and these findings were confirmed during the stakeholders' interviews. New text has been added to clarify.

c) On Step 3 - one of the questions I had when I read this section is "how many of the 200 sampled respondents responded to the survey?" I see that this comes in the Results section and am not that's the right place.

148 of the 200 eligible participants from the STRIPE countries responded to the survey – and we had this information to the results – and provided sub-headings under the results section to correspond to the methods section for better organization.

d) On Step 3 - I hope other reviewers of this paper assess the quality and relevance of the quantitative analysis done as I don't have the expertise to comment on this analysis

2. Results section:

a) I think the content on page 13 fits better in your Methods section than your Results section.

We have retained the descriptive statistics on the survey in the results section to distinguish these findings from the process of conducting the survey (which has been retained in methods section) – and to emphasize it as an outcome of the application of our research method. However, we have

provided sub-headings to organize the results section to correspond with the methods section.

b) Results section could benefit from sub-headings

Subheadings have been added

3. Discussion section:

a) I am struggling with having the 5 factors presented in this section - why not present these factors in the Results section, and use the Discussion to present the implications of the tool, and perhaps how different this tool is from tools used in HICs? It seems to me that the content on pages 25, 26 and 27 could well fit in the Results section.

We have provided an initial description and definition of these 5 factors under the results section (see page #22-23) – as a direct product of the exploratory factor analysis. However, naming and further operational definition of the factors themselves are part of our own interpretation and so we felt it was best to reflect on this thinking in the discussion, rather than in the results. The results section here is designed to display how we reached the five distinct factors, rather than our interpretation of them.

b) On page 28, please provide the reference for the statement made in line 5 and 6.

Thank you, this has now been added.

Finally, the paper may benefit from another read to pick out a few typos, e.g. on page 21 line 11.

The specific error you noted has been fixed and we've also done a re-read to identify other typos.

Thank you again for your review.